# A Practical Approach for Uncertainty Management in Rubber Manufacturing Processes Using Physics-Informed Real-Time Models

**DOI:** 10.3390/polym14102049

**Published:** 2022-05-17

**Authors:** Ismael Viejo, Salvador Izquierdo, Ignacio Conde, Valentina Zambrano, Noelia Alcalá, Leticia A. Gracia

**Affiliations:** Instituto Tecnológico de Aragón (ITAINNOVA), Calle María de Luna 7, 50018 Zaragoza, Spain; iconde@itainnova.es (I.C.); vzambrano@itainnova.es (V.Z.); nalcala@itainnova.es (N.A.); lgracia@itainnova.es (L.A.G.)

**Keywords:** process modeling, rubber industry, material characterization, finite element analysis, uncertainty

## Abstract

Industrial manufacturing management can benefit from the use of modeling. For a correct representation of the manufacturing process and the subsequent management, the models must incorporate the effect of the uncertainty propagation throughout the stages considered. In this paper, the proposed methodology for uncertainty management uses a nonintrusive method that is based on building a deterministic physics-informed real-time model for the a posteriori computation of output uncertainties. This model is built using tensor factorization as the Model Order Reduction technique. It includes as model parameters: material properties, process operations, and those random and epistemic uncertainties of known variables. The resulting model is used off-line to identify sensitivities and therefore to unify uncertainty management across the material transformation process. This method is presented by its direct application to an automotive door seal manufactured by continuous co-extrusion of several rubbers and reinforcement (metal strip and glass fiber thread).

## 1. Introduction

Product manufacturing activities are typically performed in an uncertain environment. There are external and internal disturbances in the planned production processes, which companies must react to in order to remain competitive in the market and guarantee a product without defects, making manufacturing systems more efficient [1]. Traditionally, uncertainty management in manufacturing lines was based on the knowledge of qualified employees; however, in the present, computer tools and automated factories have facilitated the data collection, the data interpretation, and the generation of decisions to detect unexpected events and initiate corrective actions [2]. However, uncertainties propagate through the whole design cycle from material characterization to material forming and modeling approaches need to capture this propagation.

Uncertainty Quantification (UQ) is the branch of knowledge that characterizes and reduces uncertainties in both physical and computational applications, trying to quantify how some results or measurements are affected if some aspects of the system are different, even slightly different. Therefore, it requires collecting enough data. The utilization of data-driven approaches such as physical models has increased [3,4,5], with it being feasible to generate big datasets for any combination of input parameters. Generating physical models are expensive, but once they are built, they provide an enormous amount of data without noise and the outputs available have no limitations as when using measured quantities. Additionally, to perform UQ analysis, the physical models provide other capabilities such as understanding the system by means of execution of the virtual factory.

There are two criteria for classification of the existing physics-based UQ and propagation approaches [6,7]: (i) one based on how intrusive they are in the simulation models, and are therefore named normally as the intrusive or nonintrusive approach; and (ii) one based on the completeness of the simulation approach, such as full modeling on one side or subrogate modeling on the other side. The application of UQ for industrial applications can be approached differently based on previous classifications. Normally, these industrial applications are very dependent cases, with the applications of intrusive methods being difficult and, thus, limiting the generalization and the robustness of their implementation. On the other hand, the time required for full modeling substantially limits their use in shop floor deployments.

Recently, UQ methods based on highly accurate subrogate modeling have been developed. The approaches utilized are mainly intrusive methods, i.e., [8,9]. These approaches rely on subrogate models using techniques such as Reduced Order Methods (ROM), which provide the uncertainty evaluation by means of a real-time physics-based simulation tool.

In this paper, we introduce a nonintrusive UQ method based on ROMs, and specifically, tensor factorization techniques [10]. The nonintrusive ROM-based UQ approach is applied to a specific industrial manufacturing test case, viz. an automotive sealing profile manufactured by the co-extrusion process. The approach to introduce the method focuses on the application. We believe that, in this way, the methodology is properly demonstrated and can be used as a user’s guide for further implementations, as well.

With the purpose of providing the UQ approach with this application, a full simulation model is required in the generation of the subrogate model. In this way, the simulation of rubber extrusion processes has been presented by several authors, i.e., [11,12,13]. However, this case presents, as a critical point, the modeling of the curing/foaming kinetics of the rubber. The coupling of both kinetics with the thermal/mechanical processes is very relevant and becomes a novelty. The curing process (aka vulcanization) changes the molecular structure of the rubber, avoiding the ability to undergo plastic deformation [14]. This process has been studied from an experimental point of view [14,15] and also from a numerical perspective [16]. Significant knowledge of the evolution of the curing and foaming process of the rubber-foam product and the physical and mechanical changes along the production line is required to simulate the process and reduce manufacturing defects. The proposed full simulation model aims to reproduce the curing/foaming process through several ovens, where the coupling of thermal, mechanical, and kinetic fields becomes of crucial importance.

Therefore, this paper presents as a novelty the way in which UQ is managed and implemented in an industrial case, as well as the development of the modeling of such an industrial case.

Section 2 describes the methodology for the UQ and generation of the physics-informed model, which is based on the Finite Element Method (FEM) for the simulation of the curing and foaming process of a co-extruded seal. With the purpose of modeling this process, the coupling of thermal, mechanical, and kinetic fields becomes fundamental. In Section 3, it is described how to use this model to build a real-time simulation approach that allows managing uncertainty. Finally, Section 4 collects the conclusions focusing on the issues of this deployment and possible future applications for this approach.

## 2. Methods

Our objective is to provide a methodology to be easily deployed in industrial environments. Two main requirements for the methodology are: (i) it must be adaptable to a wide range of simulation approaches, such as the dynamic system simulation, Finite Element Method (FEM), Computational Fluid Dynamics (CFD), or a combination of them, a requirement that leads to the necessity of a nonintrusive approach; and (ii) it must be valuable at the shop floor level, which means that sensitivity analysis as well as forward and inverse UQ studies are needed to be performed in real-time. Based on these restrictions, we propose the approach summarized in Figure 1, where the material model and the processes operation are analyzed in parallel. Aleatory uncertainties are considered by including variable parameters in the simulation models at the material and process level. Epistemic uncertainties are considered by including model parameters as global parameters. The methodology is designed so that it can be applied to any manufacturing process, although this paper tests its application on a rubber extrusion process, which itself becomes remarkably challenging from a modeling point of view.

The case studied consists in the manufacturing of an automotive sealing profile made of EPDM (Ethylene Propylene Diene Terpolymer) rubber and metallic fiber hoops. It is produced in a co-extrusion continuous production line, which can be divided in two main parts: (i) the extrusion process where the material goes through a die mold, and (ii) the curing/foaming process where the seal profile goes through several ovens; see Figure 2 The manufacturing parameters, especially those related to the curing/foaming process, present a high influence into the final shape of the profile, which is the most critical aspect to fulfil the required quality product.

The first part of the manufacturing process consists of two or three extruder screws that feed a die head mold where the different rubber compounding goes through the channels until the die section. Additionally, some reinforcement elements can be introduced through this mold to be part of the profile, such as a metal band or several glass fiber hoops. The differences between the rubber compounding are normally due to the inclusion or non-inclusion of foaming agents. This part of the manufacturing process, which has been extensively studied in the literature, is not the objective of this paper.

The second part of the process starts from the end of the die and consists of some thermal treatments: (i) infrared oven, (ii) microwave oven, (iii) gas convection oven, (iv) cooling bath, and (v) second gas convection oven. Between these treatments, the profiles are exposed to ambient conditions (convection with the air in the factory). During these treatments, the curing and foaming processes are activated and, once they are completed, the final shape of the product is reached. The synchronization of these curing and foaming processes is critical to reach the desired final shape of the profile.

As it has been described previously, the modeling of this extrusion process requires, as an important part, the coupling between the kinetics, thermal, and mechanical fields. This coupling is made by means of a fully coupled thermal-stress FEM analysis combined with specific subroutines developed by the authors, all of them implemented in the commercial finite element code ABAQUS [17]. These subroutines are mainly necessary to couple the kinematic field with the thermal one, functionality, which is not directly available in the commercial software. The expansion coefficient associated with the foaming degree provides the coupling with the mechanical field. Additionally, the mechanical and thermal properties show dependence on temperature and the cured and foamed state, so all the fields are coupled with the material properties evolution during the process. In order to feed the model, an experimental methodology was developed and is completely documented in [18].

The following subsections describe the implementation of the macroscopic-coupled model.

### 2.1. Mechanical Field

Vulcanized rubber behavior can be ideally assimilated to hyperelasticity; however, uncured rubber provides a behavior with plastic deformation, as described by Restrepo-Zapata [14]. We assume a linear elastic behavior for the material, as the strain level it undergoes along the process is low enough to keep the hypothesis valid at its uncured and cured states. The constitutive model also includes the expansion due to the thermal and foaming fields. The expansion due to the foaming mechanism provides the coupling between the mechanical field and foaming process.

Under previous considerations, the constitutive model for the isotropic solid with a three-dimensional stress state is expressed as:
*ε_i,j_* = *σ_i,j_* (1 + *ν*)/*E* − *σ*_k,k_
*δ_i,j_*
*ν*/*E* + *CTE* Δ*T*
*δ_i,j_* + *CFE* Δ*β*
*δ_i,j_*,(1)

where subscripts *i,j* can take values 1, 2, or 3, *σ* is stress, *ε* is strain, *E* is Young’s modulus, *ν* is Poisson’s ratio, Δ*T* is the increase in temperature of the solid, *CTE* is the coefficient of linear thermal expansion, Δ*β* is the increase in foaming degree, and *CFE* is the coefficient of linear foaming expansion that is reached when the foaming grade is unity.

The Young’s modulus depends on the curing and foaming degree. Equation (2) defines the curing dependence, which was fitted from experimental tests [18].
*E*_rubber_ (at *ε* = 10%) = f(α) = a α^2^ + b α + c
(2)

where a, b, and c are fitted parameters of the regression and α is the curing degree.

The foaming dependence is introduced based on the Mori–Tananka (M–T) approach, but particularized for spherical voids [19], and it is assumed that the standard relation for the isotropic linear elastic material between the elastic modulus and the shear and bulk modulus is valid for the porous composite [18]. These equations are introduced as a tabular dependence in FEM code.

### 2.2. Thermal Field

Equation (3) gives the three-dimensional transient heat transfer equation for an isotropic material.
*ρ C_p_**∂T*/*∂t* = *∂*/*∂x* (*λ ∂T*/*∂x*) + *∂*/*∂y* (*λ ∂T*/*∂y*) + *∂*/*∂z* (*λ ∂T*/*∂z*) + *Q*′
(3)

where (*x, y, z*) is the Cartesian coordinate system, *T* and *t* are temperature and time, respectively, *ρ* is density, *C_p_* is the specific heat, λ is the thermal conductivity, and *Q* is the heat generation rate per volume-unit.

The specific heat depends on the temperature and the curing degree with the relation established in Equations (4)–(7), but the thermal conductivity depends only on the temperature, Equation (8).

p = 1 − 1/(1 + *CFE* β)^3^
(4)


C_p_^β^ = (C_p_^α^ *ρ*_α_ (1 − p) + C_p_^air^ *ρ*_air_ p)/(*ρ*_α_ (1 − p) + *ρ*_air_ p)
(5)


C_p_^α^ = (1 − α) (a_1_ + b_1_ T)+ α (a_2_ + b_2_ T)
(6)


λ^β^ = λ^α^ (1 − p)+ λ^air^ p
(7)


λ^α^ = c_1_ + d_1_ T
(8)

where p is the porosity level, the superscripts α and β correspond with the property value of the rubber with a certain curing/foaming degree, respectively; a_1_, b_1_, a_2_, b_2_, c_1_, and d_1_ are fitted parameters of each material property, which were fitted from experimental tests [18].

### 2.3. Curing Model

Over the years, several kinematic models have been developed to fit the curing behavior of rubber components. For this paper, the phenomenological Kamal–Sourour approach [20] with an additional parameter to lock the maximum reachable value is used. The curing kinematic equation is defined as:
*∂*α/*∂*t = (K_1α_ + K_2α_ α^m^) (B_α_ − α)^n^,
(9)


K = A e^−E/T^
(10)

where α is the vulcanization degree, m and n are fitted parameters related with the reaction order, B_α_ is the lock of the maximum value, and Ks is an Arrhenius-type temperature dependence that is a function of two parameters A and E (activation energy).

The vulcanization reaction is exothermic [21,22], and therefore there is heat generation depending on the curing degree, which is modeled as:*Q*′ = *Q_total_*
*∂*α/*∂*t,
(11)

where *Q_total_* is the total heat of the reaction associated with the weight concentration of “rubber” without foaming additives, also called blowing agent (BA).

### 2.4. Foaming Model

The foaming kinetics defines how the rubber is foamed due to the bubble generation and growing. In this paper, this kinetics is also defined according to the modified Kamal–Sourour model. Additionally, this reaction equation is adapted to include the dependence with the curing degree by means of a function, called g, with an initial expression defined as Equation (12).

g(α) = e^−αh^
(12)


Hence, the foaming equation is defined as:*∂*β/*∂*t = (K_1β_ + K_2β_ β^m^) (B_β_ − β)^n^ g(α),
(13)

where β is the curing degree, and m, n, B, and Ks are defined as in Equation (10).

The foaming process is an exothermic reaction, where the heat generation rate is defined by:
*Q*′ = *Q_total_
**∂*β/*∂*t,
(14)

where *Q_total_* is the total heat of the reaction associated with the weight concentration of BA.

### 2.5. Parameter Calibration of the Coupled Model

The calibration of the constants for Equations (2)–(14) and the material properties dependences are performed based on the experimental characterization, which is described in reference [18]. The values obtained from such a material-model-parameter calibration are used in this paper.

### 2.6. Specific Subroutines

Equations (1) and (3) are directly solved in ABAQUS by using a fully coupled thermal-stress analysis. However, this ABAQUS model does not allow the solving of kinematic reactions and other features in a direct way; therefore, they must be defined by the user by using specific subroutines. The required subroutines to include the complete material behavior described in previous subsections are detailed next and summarized in Figure 3.

*Subroutine HETVAL:* This subroutine defines a heat flux due to internal heat generation in a material such as phase change. As it is already mentioned, curing and foaming kinetics are exothermic reactions that generate internal heat (see Equations (11) and (14)). This internal heat depends on the curing and foaming degree, respectively, and it is solved by the corresponding reaction rate of vulcanization (Equation (9)) and foaming (Equation (13)), through two auxiliary subroutines, f and rk4.

*Subroutine rk4:* This solves the reaction equations, by a 4th order Runge–Kuta method [23].

*Subroutine f:* This is an auxiliary subroutine to define the reaction equations.

*Subroutine USDFLD:* This subroutine allows the definition of field variables at a material point, with these field variables being available for use for other material properties in its dependence definition. For this model, two *USER FIELDS* are defined corresponding with vulcanization (*α*) and foaming (*β*) degree. These fields are used to visualize results and as input for the material look-up tables (properties with vulcanization and foaming dependence such as the Young’s modulus and thermal conductivity).

*Subroutine UEXPAN:* This subroutine defines the incremental thermal strains as a function of temperature, predefined field variables or state variables. In this model, two UEXPAN contributions are defined: (i) one corresponding with the incremental strains due to the temperature (CTE) and the (ii) other corresponding with the incremental strains due to the foaming degree (CFE).

*Subroutine FILM:* This subroutine defines a nonuniform film coefficient for thermal boundary conditions. In this model, the film coefficient is defined as a function of the velocity of the extrusion line and the temperature of each solved point. This coefficient is calculated using the correlation for laminar flow over an isothermal plate presented by Incropera [24].

### 2.7. FE Model of the Seal Profile

The seal profile is a 3D geometry with a length of about 90 m in the manufacturing process; however, its section (2D geometry) is constant along the profile length except for the changes produced due to the heating process and kinetics coupling, from this point of view, a 2D simulation, assuming plane strain could be suitable. However, in the foaming zone of the sealing, an initial pre-strain state is generated due to the screw extruder setup that compromises the use of a 2D model. Instead, a 2.5D model is defined, where the longitudinal dimension is very small and boundary conditions equivalent to a plane strain state are defined. In order to consider the different conditions along the length of the seal profile, all boundary conditions are a function of time.

The 2D geometry of the seal profile is built based on the nominal section of the die head mold; see Figure 2. Two differentiated rubber zones, coming from the different screw extruders, are considered, each one with different materials: rubber and foaming rubber; see Figure 4. In addition to these rubber parts, the geometry includes a metal band, several glass fibers, and an auxiliary geometry called a roll surface. The last one is a slider used to apply a vertical displacement to the profile at the exit of the die mold. This geometry is meshed with 8-node-coupled temperature-displacement elements (C3D8RT).

The simulation of the continuous seal extrusion process requires that the boundary conditions change along the simulation time, corresponding with the movement of the seal in the manufacturing line. The manufacturing of the profile is made by using a sequence of different heating (ovens) and cooling (ambient conditions or forced convection) sources; see Figure 5. During the process, the seal profile is suspended or rests on several supports and the metal band is co-extruded with the seal at a certain velocity. These mechanical restrictions are substituted by equivalent conditions in the model. The metal band is considered to be fixed while the seal moves through it, that is, the model predicts the relative displacements with respect to the metal band.

The simulation is divided into 11 steps corresponding with the 11 subprocess that are shown in Figure 5. The specific thermal and mechanical boundary conditions are described in the scheme of Figure 5. In addition to that description, in the first step, a pre-strain state is applied, which is associated for the screw extrusion process with a reduction in the mass flow rate of foamed rubber against the theoretical. This pre-strain is calculated based on the nominal section. This correction is introduced to compensate the longitudinal expansion produced by the foaming process. This fact is defined in the numerical model by applying a longitudinal displacement over the nodes of one of the sides of the foam rubber in the first step of the simulation.

Based on the previous FE model definition, the manufacturing parameters, which have a higher effect on the final shape of the profile, are selected. They are summarized in Table 1. Some of the manufacturing parameters, such as the power of each IR lamp of the process (there are 16 lamps in the oven), are grouped together in a unique parameter called the ratio of nominal heat in IR oven. This parameter controls a global factor of the nominal power IR lamp distribution, with 1.0 being the nominal value of the lamps. The parameter Ratio of RPM is related to the pre-strain commented previously.

These parameters are used for the ROM generation that is described in following sections.

## 3. Results and Discussion

With the purpose of validating the coupled model over the seal, this afore-described FE model is evaluated for known manufacturing values. The predicted deformed shape of the profile is compared with a digital image of the manufactured profile; see Figure 6. The comparison is rated as good if profile thickness at positions 1 to 4 lies within a tolerance. The predicted geometry shows some similarities, but there are still some zones with outstanding differences. Part of the nonconformities of both images comes from the aleatory uncertainties that are studied using the real-time model, while the others are epistemic and come from some simplifications in the FE model. Among the latest ones, some remarkable are, e.g., the way of simulating the profile closure, which is simulated in a very simplified way due to the high computational cost, or the modeling simplification of some material phenomena such as the nonconsideration of permanent strains on the rubber.

The approach proposed includes the generation of subrogates models with the purpose of uncertainty management. These subrogate models are built using a tensor factorization technique, called TWINKLE library [25]. This library allows the reduction in nonstructured and sparse datasets such as those generated by series of numerical experiments generated using Design of Experiment (DoE) approaches. The main advantage of this approach is that the resulting real-time subrogate models can be exploited in several ways. They can be used, for example, to explore visually how the material or the process reacts under changes in input parameters. Alternatively, they can be used to be explored in a stochastic way to perform forward and inverse UQ analysis with extremely low computational cost. The next subsections describe the developed real-time visualization tools and a forward UQ application to illustrate the approach.

### 3.1. Modeling Visualization

With the purpose of analyzing the uncertainties of this model, two numerical tools have been built.

(i) At the material level, the ODE system described in previous sections is solved. An interactive tool is built and used for analyzing the sensitivity of the material parameters, especially the ones regarding the coupling between the two kinetics reactions. On this basis, the aleatory uncertainties come from the material and the epistemic uncertainties come from coupling of the ODE system; and both are included as global parameters. We conclude that there is a parameter that can jointly represent both aleatory and epistemic uncertainties. This parameter is the one associated with the coefficient of foaming expansion (CFE), as it is, at the same time, a characteristic of the material that is measured in the laboratory, and the parameter used to couple different physics within the model. The reduction to a unique parameter provides a way to reduce the complexity of uncertainty management, but maintaining a high sensitivity to the material behavior.

(ii) At process level, the tool developed (see Figure 7) provides visualization of the ROM results with the purpose of understanding the behavior of the seal. That tool includes, at the left, several sliders to modify the parameter described in Table 1 together with the position in the manufacturing line that is being evaluated. The evaluation of the selected point in the ROM model is represented as a contour distribution in terms of some of the fields (e.g., temperature, vulcanization or foaming degrees, and displacement) and also a chart with the history results for a specific point of the geometry.

Therefore, the tool allows visual identification of control parameters for providing the desired final geometry. It becomes a very useful tool for both designers, as it allows the increase in their knowledge when developing new seals, as well as for manufacturing engineers, as they can learn about the parameters to modify in order to obtain the desired final product. One of the key advantages of the tool is that it provides results in real time.

The utilization of this type of tools provides a faster learning of the behavior at the two levels (material and component) for any worker in the factory, regardless of its background.

### 3.2. Uncertainty Quantification Using ROM

A UQ approach is proposed and tested on a real application case. It is assumed that input parameters of the process, i.e., velocity and the ratio of power used in the microwave oven, show a Gaussian distribution. In addition, the CFE, which relates the coupling between the vulcanization and the foaming processes within the rubber, is assumed to show a Gaussian distribution as well. A numerical DoE is generated with the following criteria: (i) a vector with random values following the defined distribution is created for each one of the three parameters considered in the UQ (velocity, microwave regulation, and CFE) and (ii) fixing the rest of the parameters of the subrogate model to the nominal value in Table 1. A histogram with the distribution of values for the three parameters analyzed in UQ is represented in Figure 8 first row. The subrogate process model predicts the results for the DoE generated, representing probability distribution functions for quality parameters of the product (Figure 8 s row). These parameters are the vulcanization and foaming degree at the end of the line; maximum, minimum, and mean temperature; and thickness of control quality positions.

This approach has been used to perform a more complete study considering additional parameters and individual effects. The main result from this complementary study is a detailed sensitivity analysis between process parameters and quality outputs [26].

The described results can be translated to the shop floor in two complementary ways: The first one is related to people as real-time simulation tools allow the acceleration of the learning processes and improving of vertical and horizontal integration within the company. This is emphasized when UQ is added as additional information. The second way is related to quality control, as the forward UQ approach allows an a priori six-sigma quality control strategy instead of the classical a posteriori control.

## 4. Conclusions

This paper presents a practical nonintrusive method for uncertainty management that can be applied to continuous manufacturing processes. This method is based on the utilization of real-time subrogate models based on numerical simulation such as FEM. The approach is applied to an automotive product: a seal profile for car door/body, which is produced by a co-extrusion process, where a complex mechanism with multiple interactions (mechanical, thermal, and kinetics) takes place and provides the final shape of the product, with the final shape being the criteria to assure the quality of the product. The method section includes a detailed description of the simulation model that illustrates the complexity and the interaction of theses mechanisms.

The approach described in this paper presents as main contributions: (i) the simulation of the material and processes under an nonintrusive approach; (ii) the use of a subrogate model generated by using the tensor factorization ROM approach; (iii) that the real-time ROM facilitates the role of visualization and interactive result exploration; (iv) the ROM evaluation requires very low computational resources and time, all facilitating the forward and inverse UQ applications even for on-line evaluation.

Those novelties, especially the non-intrusive approach, emphasizes the straightforward transference and application to any manufacturing processes that can be generated by a full simulation of the process, such as an injection molding process, for which there are several software available for the simulation.

The future improvement for this approach, aligned with our focus, should be the development of a dynamic real-time simulation, based on the presented steady-state strategy, to support a model-based control strategy under uncertainty.

## Figures and Tables

**Figure 1 polymers-14-02049-f001:**
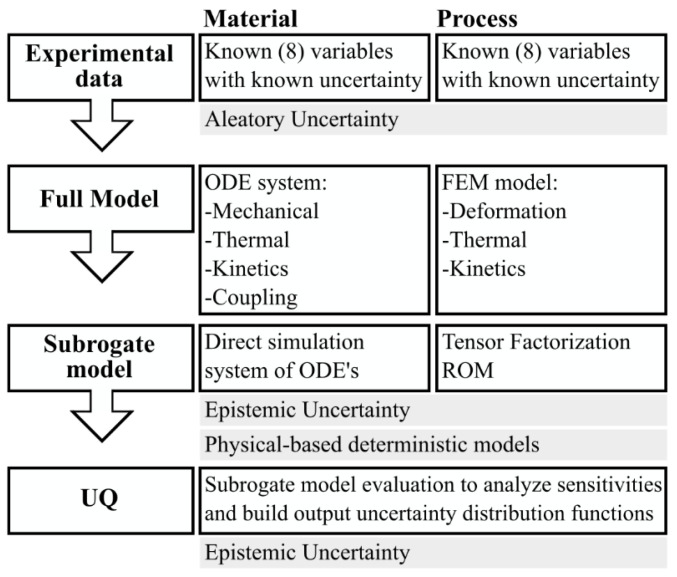
Summary of the uncertainty management methodology with application to material transformation processes.

**Figure 2 polymers-14-02049-f002:**
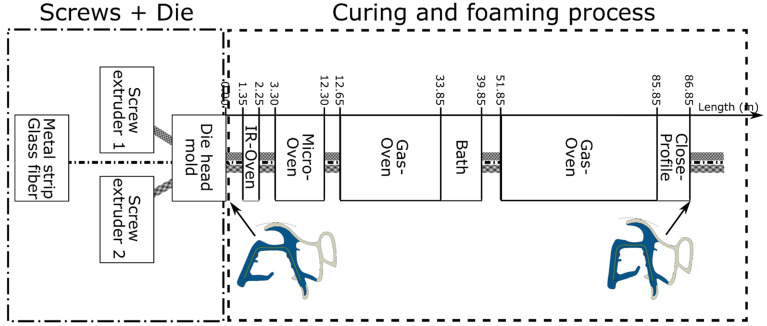
Scheme of the manufacturing process for the automotive door seal.

**Figure 3 polymers-14-02049-f003:**
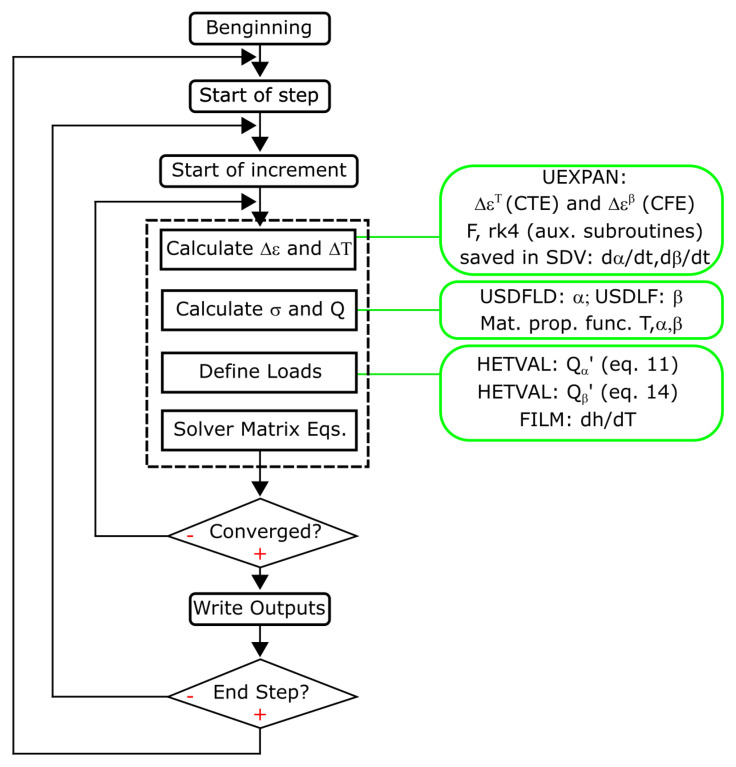
Fully-coupled algorithm of thermal-mechanical simulation with subroutines for kinetics coupling.

**Figure 4 polymers-14-02049-f004:**
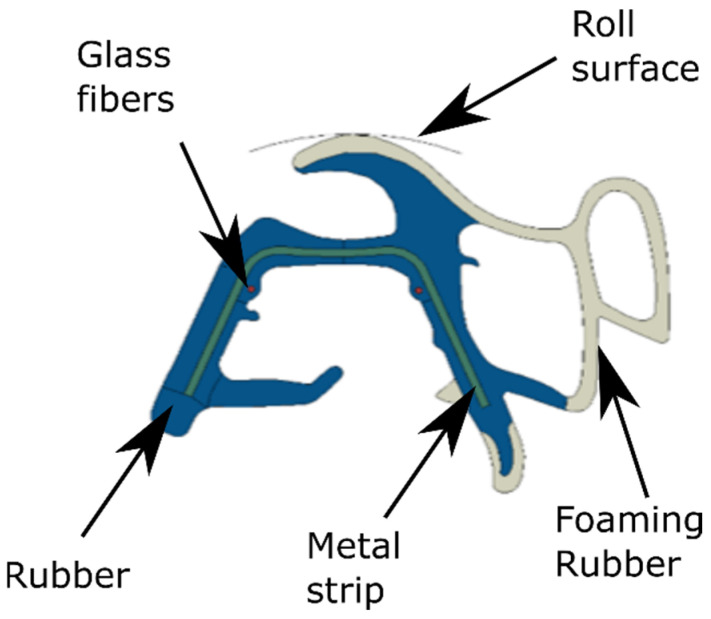
Parts of the automotive door seal.

**Figure 5 polymers-14-02049-f005:**
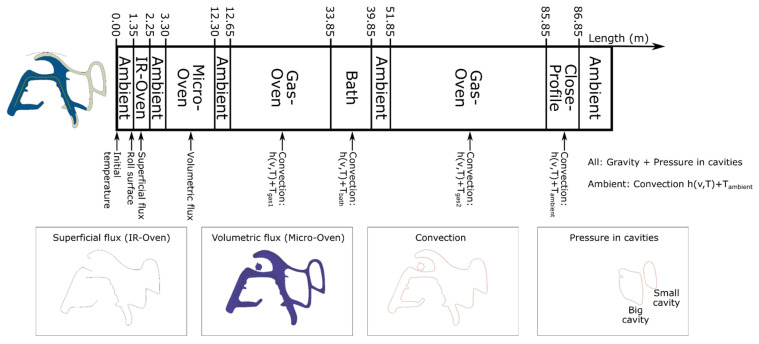
Scheme of the thermal treatment of the extrusion line and associated boundary conditions applied for the simulation.

**Figure 6 polymers-14-02049-f006:**
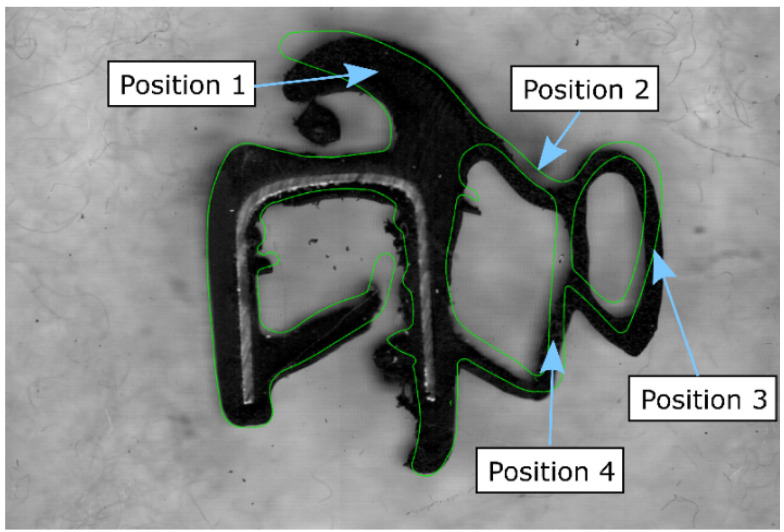
Position where the thickness is measured also includes the overlaying of the manufactured product (black section) and the FEM results (green line).

**Figure 7 polymers-14-02049-f007:**
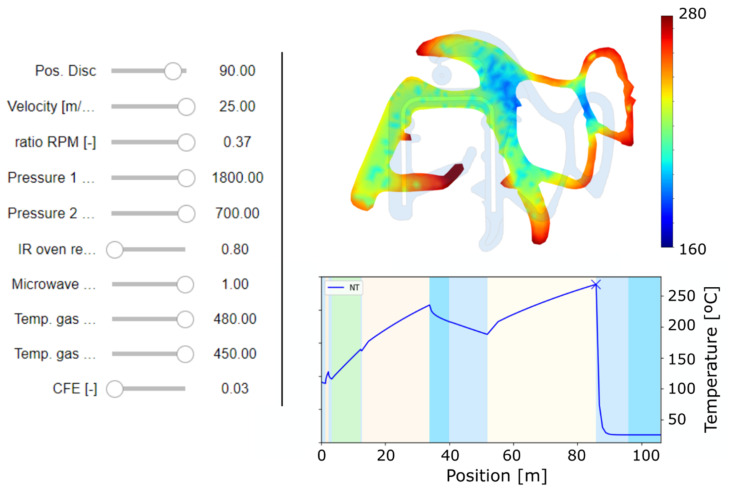
Numerical tool to visualize and explore results from the process subrogate model.

**Figure 8 polymers-14-02049-f008:**
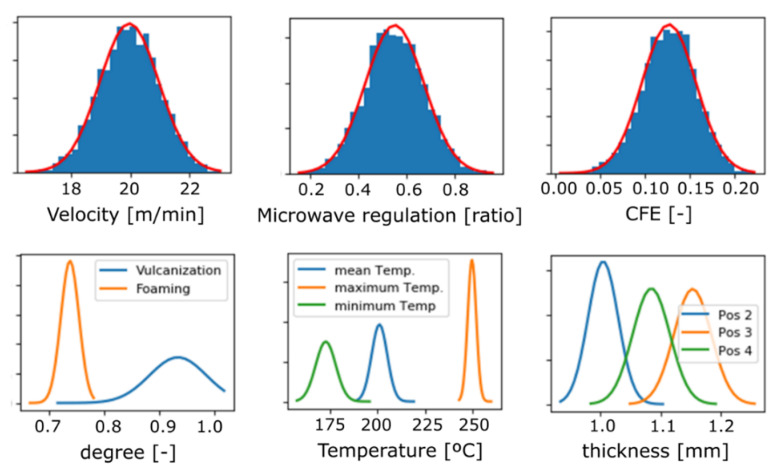
Forward UQ application using process subrogate modeling.

**Table 1 polymers-14-02049-t001:** Parameters of the FE model.

Parameter	Type of Parameter	Value ± Range
Velocity of pulling the profile	uncertainty	20 ± 5 m/min
Pressure in big cavity	controllable	1500 ± 300 Pa
Pressure in small cavity	controllable	400 ± 300 Pa
Ratio of RPM foamed vs. no foamed	controllable	0.335 ± 10%
Ratio of nominal heat in IR oven	controllable	0.95 ± 0.15
Ratio of nominal heat in Microwave oven	uncertainty	0.55 ± 0.45
Temperate in Gas1 oven	controllable	380 ± 100 °C
Temperate in Gas2 oven	controllable	350 ± 100 °C
Coefficient of foaming expansion	uncertainty	0.1275 ± 0.1025

## Data Availability

Not applicable.

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
