# Peer review of "A Practical Approach for Uncertainty Management in Rubber Manufacturing Processes Using Physics-Informed Real-Time Models"

_polymers, 2022, doi:10.3390/polym14102049_

Round 1

Reviewer 1 Report

In the paper, the uncertainty management in rubber processing is discussed.  

The paper is interesting and valuable, however, it is not clearly written.

Uncertainty management is far from the science of polymers. Therefore, some basic information on this subject should be introduced into the paper to be more understandable for readers of Polymers.

The manufacturing process is poorly described:

  • lack of a good process scheme (Figure),
  • lack of Figure of the manufactured product,
  • process parameters are not clearly described (Table 1), e.g. big/small cavity, ratio of nominal heat.

Equations are not clearly described. Check all of them. Many parameters are not explained.

The simulations are poorly described, e.g. lack of material data, rheological, thermal etc.

The authors write: “The fitting of the constants for equations (2) to (14) and the material properties dependences are performed based on the experimental characterization, which is described [22]. The values obtained for the material-model-parameter fitting are included in [22]”. However, [22] is unpublished !!!.

Subrogate models ?

The literature is not very up-to-date.

The research is described very generally (vaguely) and it is difficult to assess its quality.

Author Response

Point 1: The paper is interesting and valuable, however, it is not clearly

written.

Response 1: Several details have added in order to get a better compreshion

Point 2: Uncertainty management is far from the science of polymers.

Therefore, some basic information on this subject should be

introduced into the paper to be more understandable for readers

of Polymers.

Response 2: Additional explanation of UM have included in the introduction

Point 3: The manufacturing process is poorly described:

lack of a good process scheme (Figure),

lack of Figure of the manufactured product,

process parameters are not clearly described (Table 1), e.g.

big/small cavity, ratio of nominal heat.

Response 3:

A new figure (Figure 2) has been added explained the process, also more detail in text is included

The manufactured product was in Figure 5

Additional explanation about the parameters in Table 1 have added, just before Table 1.

Point 4: Equations are not clearly described. Check all of them. Many

parameters are not explained.

Response 4: All equation have been double checked and included some missed explanation about some paramenters

Point 5: The simulations are poorly described, e.g. lack of material data,

rheological, thermal etc.

Response 5: More details about the simulation description have been added. The experimental data is not included because it has been especifically explained in other paper of Polymer “Alcalá, N.; Castrillón, M.; Viejo, I.; Izquierdo, S.; Gracia, L.A. Rubber Material-Model Characterization for Coupled Thermo-Mechanical Vulcanization Foaming Processes. Polymers (Basel). 2022, 14, 1101, doi:10.3390/polym14061101.”

Point 6: The authors write: “The fitting of the constants for equations (2)

to (14) and the material properties dependences are performed

based on the experimental characterization, which is described

[22]. The values obtained for the material-model-parameter

fitting are included in [22]”. However, [22] is unpublished !!!.

Response 6: It was a mistake, when this paper was in draft the reference 22 was unplublished and once that reference was published not in all reference in the paper was correctly updated.

Point 7: Subrogate models ?

Response 7: I don’t understant what do you mean with this comment. Please, could you explain in more detail.

Point 8: The research is described very generally (vaguely) and it is

difficult to assess its quality.

Response 8: Several details have added in order to get a more detailed descriptions

Reviewer 2 Report

This paper lacks important details, such for example:

  • Details and explanations about the way how random and epistemic uncertainties are introduced in the model (page 2, section 2);
  • Details about the methodology related to the previous point;
  • Figure 1 must be explained in detail;
  • Most methods referred to in section 2 must be explained in more detail, namely, sections 2.4, 2.5 and 2.6. In this last case, details about the subroutines must be provided.

The absence of details does not allow to the reader understand and reproduce the work. Also, it was not clear the distinction between what belongs to the commercial software used and what is done by the authors.

The model proposed only was assessed using a single example. Seems that this does not allow to conclude about the generalization of the methodology proposed, as the authors referred to at beginning of the text, in the introduction.

Author Response

Point 1: Details and explanations about the way how random andepistemic uncertainties are introduced in the model (page 2,section 2);

Response 1: We think that the explanation given in section 3.1 could help for compression how they are introduced

Point 2: Details about the methodology related to the previous point;

Response 2: As it is described the uncertainties are considered as parameters of the model that are utilized on the generation of the subrogate model, and so, they can utilized for the UQ evaluation.

Point 3: Figure 1 must be explained in detail;

Response 3: It is decribed in first parragraph of section 2

Point 4: Most methods referred to in section 2 must be explained in moredetail, namely, sections 2.4, 2.5 and 2.6. In this last case, details about the subroutines must be provided.

Response 4: Several details have added in order to get a better compreshion

Point 5: The absence of details does not allow to the reader understandand reproduce the work. Also, it was not clear the distinction between what belongs to the commercial software used and what is done by the authors.

Response 5: Several details have added in order to get a better compreshion in that sense

Point 6: The model proposed only was assessed using a single example. Seems that this does not allow to conclude about the generalization of the methodology proposed, as the authors referred to at beginning of the text, in the introduction.

Response 6: It has been applied for other applications unpublished.

Round 2

Reviewer 1 Report

The paper has been improved. However, some Figures are hardly legible, e.g. Figure 7 and Figure 8.

Author Response

Point 1: The paper has been improved. However, some Figures are hardly legible, e.g. Figure 7 and Figure 8.

Response 1: We have added a more detailed description of these figure inside the text of the paper. We thank that it should help to get a better compreshion

Round 3

Reviewer 1 Report

These Figures should be improved.

Round 4

Reviewer 1 Report

It is OK.